# A tight relationship between BOLD fMRI activation/deactivation and increase/decrease in single neuron responses in human association cortex

Marie-Alphée Laurent[1]*, Corentin Jacques[1], Xiaoqian Yan[2], Pauline Jurczynski[3], Sophie Colnat-Coulbois[1,4], Louis Maillard[1,5], Steven Le Cam[3], Radu Ranta[3], Benoit R Cottereau[6], Laurent Koessler[1], Jacques Jonas[1,5], Bruno Rossion[1,5]*

[1]Université de Lorraine, CNRS, IMoPA, Nancy, France; [2]Fudan University, Institute of Science and Technology for Brain-Inspired Intelligence, Shanghai, China; [3]Université de Lorraine, CNRS, CRAN, Nancy, France; [4]Université de Lorraine, CHRU-Nancy, Service de Neurochirurgie, Nancy, France; [5]Université de Lorraine, CHRU-Nancy, Service de Neurologie, Nancy, France; [6]Centre de Recherche Cerveau et Cognition, Université Toulouse 3 Paul Sabatier, CNRS, Toulouse, France

**\*For correspondence:**
marie-alphee.laurent@univ-lorraine.fr (M-AL);
bruno.rossion@univ-lorraine.fr (BR)

**Competing interest:** The authors declare that no competing interests exist.

## eLife Assessment

This **valuable** short paper is an ingenious use of clinical patient data to address an issue in imaging neuroscience. The authors clarify the role of face-selectivity in human fusiform gyrus by measuring both BOLD fMRI and depth electrode recordings in the same individuals; furthermore, by comparing responses in different brain regions in the two patients, they suggested that the suppression of blood oxygenation is associated with a decrease in local neural activity. The methods are **solid** and provide a rare dataset of potentially general importance.

**Abstract** The relationship between Blood-Oxygen-Level-Dependent (BOLD) responses in functional magnetic resonance imaging (fMRI) and increases or decreases in neural firing rate across human brain regions, especially the association cortex, remains largely unknown. Here, we contrast direct measures of neuronal activity in two adjacent brain regions of the fusiform gyrus (FG) associated with fMRI increases (lateral FG portion) or decreases (medial FG portion) of the same category-selective neural activity. In both individual brains tested across multiple recording sessions, a frequency-tagging stimulation objectively identified a substantial proportion (about 70%) of face-selective neurons. While single units recorded in the lateral FG showed a selective increase to faces, neurons localized in the medial FG decreased spiking activity selectively to faces. Beyond a relative reduction to faces compared to non-face objects, about a third of the single neurons found in the medial FG showed genuine suppression of baseline spiking activity upon presentation of a face. These observations clarify the nature of face-selective neural activity in the human brain, which can be expressed both as increases and active suppressions of spiking activity and, more generally, shed light on the physiological basis of the fMRI signal.

## Introduction

Functional magnetic resonance imaging (fMRI) has revolutionized our understanding of human brain function. While animal studies have characterized Blood-Oxygenation-Level-Dependent (BOLD) fMRI activation as a correlate of local synaptic activity (*Logothetis et al., 2001*; *Bartolo et al., 2011*), the cellular mechanisms of deactivation (negative BOLD signals) remain unknown. Since BOLD activity relies on contrasts between (two) conditions of interest, fMRI deactivations have been attributed to various causes, that is (relative) decreases in neuronal activity (*Shmuel et al., 2006*; *Devor et al., 2008*; *Boorman et al., 2010*), (relative) increases in neural activity (without concomitant compensation by cerebral blood flow [CBF] increase; *Schridde et al., 2008*), or hemodynamic contributions ('blood stealing'; *Harel et al., 2002*). However, notwithstanding recent evidence of fMRI deactivation associated with electrocorticographic alpha power decrease in the human visual cortex (*Fracasso et al., 2022*), most studies have been carried out in animal models and primary sensory or motor regions. Therefore, the relationship between BOLD responses and increases or decreases in neuronal firing rate across human brain regions, especially the association cortex, remains unknown.

Here, thanks to a unique opportunity to record fMRI and spiking activity in neighboring populations of neurons of the same region of the association cortex, we shed light on the nature of the relationship between category-selective activity recorded at macroscopic and cellular scales. Specifically, two patients with refractory epilepsy (P1, P2) were implanted with hybrid macro-microelectrodes in their fusiform gyrus, a hominoid-specific cortical structure (*Weiner and Zilles, 2016*) that is critically involved in face recognition (*Cohen et al., 2019*). While microelectrodes recording single-unit activity located in the lateral portion of P1's middle fusiform gyrus (MidFG) (i.e., in cytoarchitectonic area FG4; *Lorenz et al., 2017*), a region known to elicit larger BOLD activity to faces than non-face objects ('Fusiform Face Area', FFA; *Kanwisher et al., 1997*), in P2 they fell in the medial MidFG portion (FG3), in which lower BOLD responses to faces than objects are typically observed (*Pelphrey et al., 2003*; *Kanwisher, 2017*; *Gao et al., 2018*). By recording face-selective neural responses both in spiking activity and fMRI signals (independent sessions) in the same brains and in two neighboring regions, we tested the hypothesis of a systematic relationship between the two types of signals. In particular, we hypothesized that fMRI deactivations to faces were associated with a majority of face-selective decreases in spiking activity.

## Results

In both MidFG regions sampled, robust differential activity to natural images of faces *vs.* non-face categories was objectively recorded with a similar well-validated frequency-tagging paradigm in fMRI (*Figure 1A*) and microelectrode electrophysiological recordings (*Figure 1B*).

### fMRI category-selective responses

*Figure 2A and D* illustrate the category-selective regions identified with fMRI, on axial, coronal, and sagittal slices in both participants ($Z > 3.1$; $p < 0.001$), relative to the microelectrodes location. Fast Fourier Transform (FFT) of the BOLD response time courses is used to define face-selective voxels in the frequency domain (0.111 Hz; *Gao et al., 2018*). BOLD frequency spectra at each voxel are transformed into Z-scores, which can either be positive (higher response to face than non-face stimuli, i.e., BOLD signal increase) or negative (lower response to faces than non-face stimuli, i.e., BOLD signal decrease).

As expected, faces elicited larger BOLD responses than non-face objects in the lateral MidFG, but lower responses in the medial MidFG (*Pelphrey et al., 2003*; *Gao et al., 2018*). Microwires in P1 (estimated Talairach location −40, −46, −19; *Figure 2A*) fell in the lateral MidFG face-selective region: the Fusiform Face Area ('FFA', peak coordinate: −35, −51, −18; *Quian Quiroga et al., 2023*), while microwires in P2 (29, −41, −12; *Figure 2D*) were located in the medial MidFG, an unprecedented sampled region with lower (deactivated) BOLD signal for faces.

### Cellular category-selective responses

Recordings of category-selective spiking activity in P1 and P2 took place over up to 11 days, with single units detected on 7 microwires per session. Across all independent sessions (i.e., on different half-days), we recorded 245 units ($N = 206$ single-units, SU and $N = 39$ multi-units, MU; pooled over

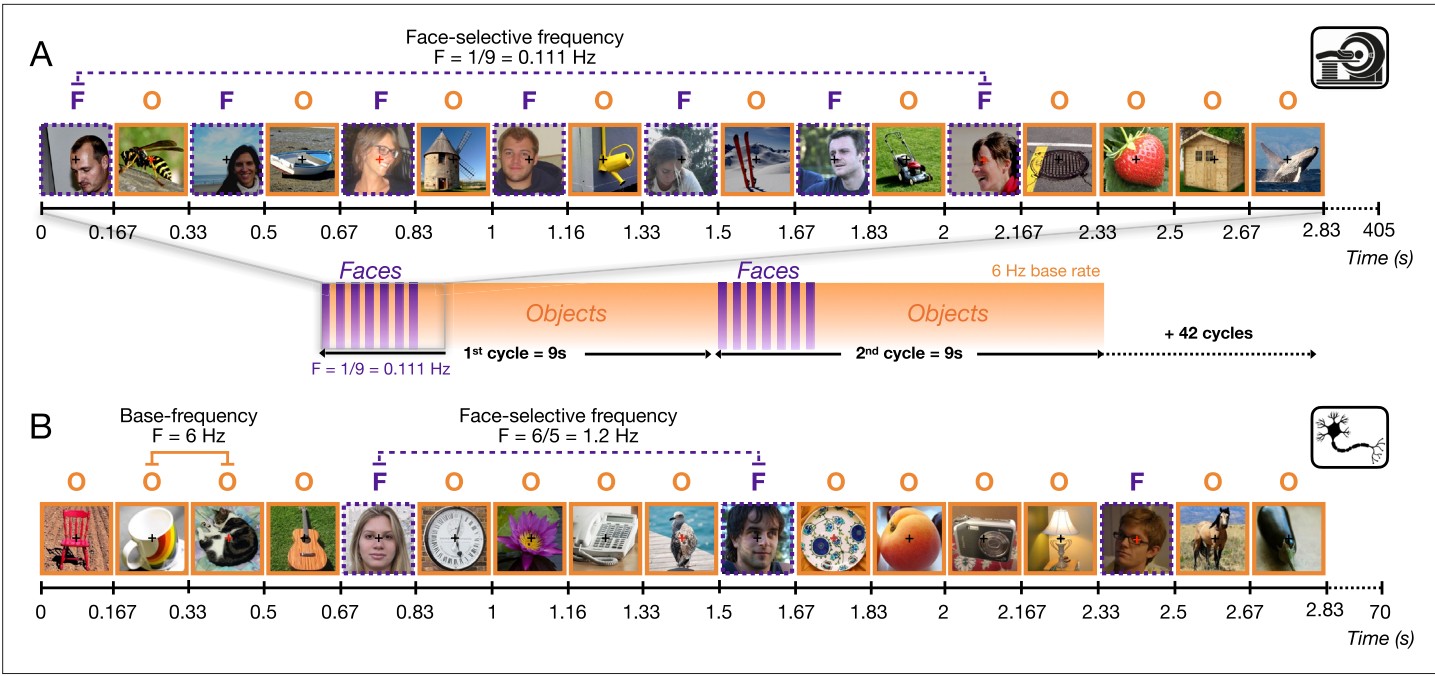

**Figure 1.** The frequency-tagging Face Localizer paradigm in fMRI (*Gao et al., 2018*) and intracerebral electrophysiological recordings (*Jonas et al., 2016*). In both cases, the same variable natural non-face images alternate at a 6 Hz rate (1 fixation/image). (**A**) During a 405 s fMRI run, a 'mini-burst' of 7 images of variable faces (purple: 'F') alternating with 6 non-face object images (orange: 'O') is presented every 9 s (0.111 Hz). A full run is composed of 44 cycles (3 s shown here). (**B**) During intracerebral recordings, variable face images are inserted periodically every fifth image (1.2 Hz). Each recording includes two 70 s stimulation sequences (3 s shown here). With both recording methods (and EEG; *Rossion et al., 2015*), this paradigm provides robust population-level face-selective activity devoid of low-level sensory confounds at the tagged frequencies. Here it is applied to single and multi-unit recording activity in the human fusiform gyrus.

4 sessions for P1 and 7 sessions for P2) firing to variable pictures of faces and objects. Following FFT, common visual responses to faces and objects were identified at the base frequency rate and harmonics (6, 12, and 18 Hz) while face-selective responses were measured at the specific face-stimulation frequency and harmonics (i.e., 1.2, 2.4, 3.6, and 4.8 Hz; *Figure 1B*; *Rossion et al., 2015*; *Jonas et al., 2016*). Despite a brief recording time for each unit (2 sequences of 70 s), only a small proportion of single neurons failed to respond (i.e., no significant base rate response: $N$ = 10/62, 16.1% in P1; $N$ = 17/144 11.8% in P2; all $Z$ < 1.64, all $p_s$ < 0.05; and no difference between participants, $p$ = 0.54, Pearson's $\chi^2$). Among all visually responsive neurons, we found a very high proportion of face-selective neurons ($p$ < 0.05) in both activated and deactivated MidFG regions (P1: 98.1%; $N$ = 51/52; P2: 86.6%; $N$ = 110/127). Even at a more conservative threshold ($p$ < 0.01), strong face-selectivity is observed (P1: 86.5%; $N$ = 45/52; P2: 77.9%; $N$ = 99/127; no difference between the two individuals, $p$ = 0.27, Pearson's $\chi^2$ test).

We then determined whether face-selective single neurons ($p$ < 0.01) recorded in activated or deactivated face-selective regions exhibited different types of responses to faces relative to non-face objects. While the vast majority of face-selective neurons in the lateral MidFG (P1) showed a significant relative increase in spike rate to faces (88.9% increases [$N$ = 40/45] $vs$ 11.1% decreases [$N$ = 5/45]; *Figure 2B and C*), most neurons in the medial MidFG (P2) showed significant relative spike rate decrease to faces (95.9% decreases [$N$ = 95/99] $vs$ 4.1% increases [$N$ = 4/99; *Figure 2E and F*). While the mean onset response latency did not differ between increases and decreases (118.2 ms $vs$. 127.3 ms, respectively; $p$ > 0.05, two-tailed percentile bootstrap; *Figure 2C and F*), a significantly lower response (i.e., absolute average firing rate) was found for decreases than increases ([130–300 ms]; 3.69 spikes/s for increases, ranging from 0.99 to 7.42 spikes/s; –2.73 spikes/s for decreases, ranging from –0.44 to –9.67 spikes/s; $t$-test $p$ < 0.01). A final noteworthy observation is that, among the 95 single units showing a reduced firing rate to faces (P2), 34 lacked a significant general visual response (at 6 Hz +12 Hz; i.e., 34.3%; $N$ = 21, 22.1% when only first 6 Hz harmonic considered, $p_s$ >0.01). Beyond a mere relative reduction to faces compared to non-face objects, responses of such

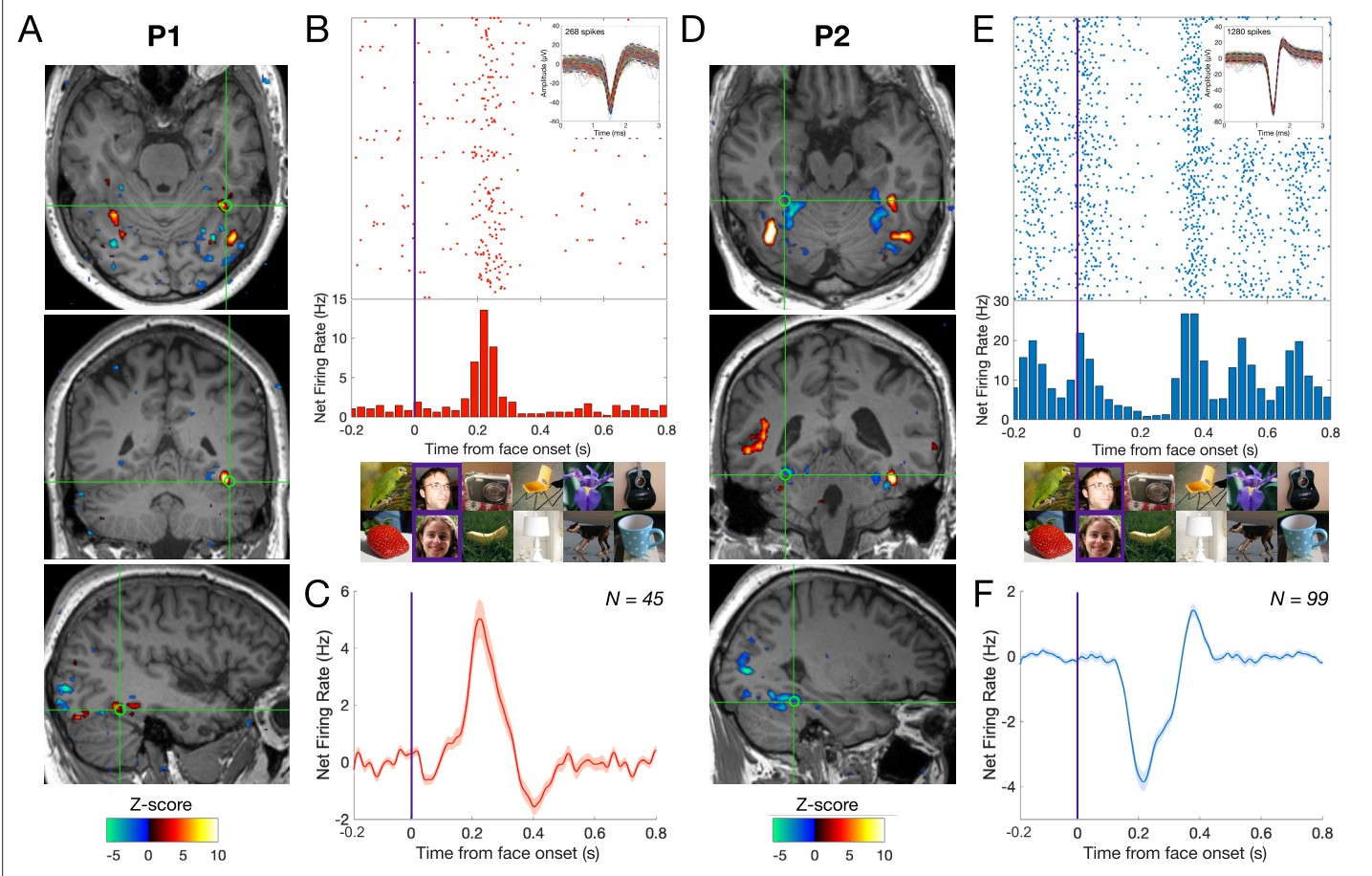

**Figure 2.** Relationship between BOLD signal and neuronal activity in the human MidFG. (**A**), (**D**) Significant face-selective activations (hot colors) and deactivations (cold colors) on axial, coronal, and sagittal slices (Z > 3.1; *p* < 0.001). The estimated location of the microelectrode (green circle) falls in left MidFG activation in P1 and right MidFG deactivation in P2. (**B**), (**E**) Representative raster plots of two face-selective single-units (SU) showing activity increase in P1 and decrease in P2 to faces, respectively. The representative face-selective SU in panel **E** also exhibits a high increase in firing rate to the general visual stimulation at 6 Hz. Each line corresponds to a 1 s epoch time-locked to the onset of a face (at 0 s), from a 140 s of recording. SU waveforms are shown in the upper-right corners. (**C**), (**F**) Average time courses of all face-selective SU identified (*N* = 45 across 4 sessions in P1, *N* = 99 across 7 sessions in P2; Z > 2.32, *p* < 0.01), showing response increase to faces in P1 and decrease to faces in P2. Note that the average time courses are computed on notch-filtered data to remove the general visual response at 6 Hz and harmonics (see Materials and methods).

'face-exclusive' single neurons therefore appear to reflect genuine suppression of baseline spiking activity upon presentation of a face.

## Discussion

Although BOLD signals correlate better with local field potentials (LFP) than spikes on a trial-by-trial basis (*Logothetis et al., 2001*), positive correlations have been observed between SU and BOLD in human (*Mukamel et al., 2005*; *Nir et al., 2007*) and non-human primates (*Goense and Logothetis, 2008*). Here, thanks to a unique combination of fMRI and microelectrode recordings in the same individual human brains, we report a striking correspondence between increases/decreases in BOLD activity and (SU) neuronal firing, in two neighboring areas of the human association cortex. By showing that the sign of the BOLD signal fluctuation during visual stimulation matches the sign of SU activity, our findings shed light on the physiology of BOLD signals, revealing in particular that BOLD decreases can be due to relative, or absolute (or a combination of both), spike suppression in the human brain. Such transient spikes suppression could be due to suppression/reduction of synaptic inputs to the recorded neuron(s) or to inhibitory inputs to the neuron(s). While differential recordings in the same hemisphere(s) and individual brains would have been optimal, our results are unlikely to

be due to differences between the two individuals or hemispheres reported, for two reasons. First, notwithstanding the right hemisphere's dominance in human face recognition (*Jonas et al., 2016*; *Kanwisher, 2017*; *Gao et al., 2018*), the same contrasted pattern of BOLD activation/deactivation for faces in the lateral/medial portions of the fusiform gyrus, respectively, has been systematically observed across hemispheres (*Pelphrey et al., 2003*; *Figure 2*). Second, while a larger number of single units was found for P2 than P1 recordings, the proportions of face-selective units did not differ between the two individuals tested.

Until now, the cellular basis of the relative BOLD decrease to faces in the medial portion of the fusiform gyrus, attributed to a smaller increase to faces relative to non-face objects or an absence of response to faces (*Kanwisher, 2017*), has remained somewhat of a mystery. Neurons showing suppression of activity to faces have been reported in the monkey inferior temporal cortex (*Salehi et al., 2020*), including inside face-selective cortical regions (*Tsao et al., 2006*; *Bell et al., 2011*). However, both the lack of reported BOLD deactivation to faces in the monkey brain and of a fusiform gyrus in non-hominoid primates *Weiner and Zilles, 2016* have prevented single-unit recordings in macaque monkeys to address this issue. Here, our neurophysiological recordings in humans suggest that faces do not only elicit a strong decrease of neuronal activity relative to non-face objects in the medial MidFG, but a genuine suppression of baseline spiking activity in a substantial proportion of the units recorded. Thus, beyond contributing to clarifying the physiological basis of the fMRI signal, our study sheds light on the nature of face-selective neural activity in the human brain, which can be expressed both as increase and active suppression of spiking activity. Inhibitory cellular activity is thought to play an important balancing role in maintaining stable function of cortical circuits (*Vogels et al., 2011*) and has been shown to play a key role in cortical category selectivity (*Wang et al., 2000*). Further direct recordings in the human association cortex will be necessary to determine the functional relevance (i.e., relationship to visual recognition behavior) and the cellular mechanisms (linked to cyto- and myelo-architecture; *Lorenz et al., 2017*) underlying these category-selective (excitatory and) inhibitory neuronal activities.

## Materials and methods

### Participants

The study included two right-handed participants (two males, aged 23 and 46) who were patients undergoing clinical intracerebral evaluation with depth electrodes (SEEG; *Talairach and Bencaud, 1973*) for refractory partial epilepsy in the Epilepsy Unit of the University Hospital of Nancy, France, in 2019 (P1) and 2023 (P2). The participants gave written informed consent to participate in the study (REUNIE, 2015-A01951-48), which was approved by a national ethical committee (CPP Est III, N°16.02.01).

### Intracerebral recordings acquisition

Electrophysiological data were recorded from Behnke-Fried hybrid micro-macro electrodes (Ad-Tech Medical Instrument). Intracerebral depth electrodes were stereotactically implanted within the participant's brains to delineate their seizure onset zone for clinical purposes. Electrode implantation sites were determined based on non-invasive data collected during the early stages of the investigation and were verified by CT scan fused with a pre-operative T1-weighted MRI. The present report focuses on recordings from the macro-microelectrodes, i.e., electrodes with macro-contacts modified to include 8 recording microwires and one reference, protruding about 3–4 mm beyond the tip of the macro-electrode (Ad-Tech Medical; see *Salado et al., 2018* for details about the macroelectrode implantation procedure). The macro-microelectrodes targeted the left (P1) and the right (P2) MidFG.

The microwires' signals were recorded at a 30 kHz sampling rate, using a 256-channel amplifier (BlackRock Microsystems), and recording activity was band-pass filtered (0.3–7500 Hz). Here, we report microelectrode data recorded during 3 days (8 sequences performed in P1) and 11 days (14 sequences performed in P2). These differences across participants were due to the clinical context in which the experiment took place. Participants were tested individually and seated at 80 cm from the computer screen.

## Microwires localization in the individual anatomy

The exact position of each contact from the micro-macro electrode relative to the individual brain anatomy was determined in each participant's brain by coregistering the post-operative CT scan with a T1-weighted MRI of the patient. The microwires' coordinates were estimated at approximately 4 mm from the tip of the macroelectrode. In addition, the location of microwires relative to fMRI-defined face-selective regions was determined by having all participants perform an fMRI face localizer a few months after the SEEG procedure, using an fMRI version of the frequency-tagging paradigm providing high SNR and reliability (*Gao et al., 2018*). After fMRI scanning, estimated microelectrode coordinates were rendered into the T1-weighted MRI co-registered and normalized to ACPC space with the functional Z-score map, using *MRIcroGL* for visualization. The location of the microelectrodes was also defined in relation to the boundaries of the cytoarchitectonic areas FG2/FG4 and FG1/FG3 (*Rosenke et al., 2018*).

## Stimuli

Stimuli were selected from a large pool of 200 to 218 non-face object images and 45 to 48 face images (as in e.g., *Jonas et al., 2016*), presented in grayscale or color sequences. Each image contained an unsegmented object or face near the center, varying significantly in size, viewpoint, lighting conditions, and background (*Figure 1*). Images were equalized for mean pixel luminance and contrast, but low-level visual cues associated with the two categories (faces and objects) remained highly variable, naturally minimizing the contribution of low-level visual cues to the face-selective neural responses (*Rossion et al., 2015*).

## Face localizer paradigms

A well-validated Fast Periodic Visual Stimulation (FPVS) face localizer paradigm was performed to define face-selective neural activity. Highly variable natural images of faces were embedded periodically within a rapid 6 Hz stream of object images. In the fMRI version (from *Gao et al., 2018*; also, *Laurent et al., 2023*), mini-face blocks ('bursts') were presented for 2.167 s every 9 s to account for the sluggishness of the BOLD response (i.e., $F = 1/9 = 0.111$ Hz). Each mini-block consisted of a set of seven face images alternated with six non-face object images to avoid category adaptation and maximize the contrast between faces and objects (*Figure 1A*). Each fMRI sequence lasted 405 s and contained 44 cycles of face bursts (including a 4.5 s baseline at both the beginning and end of each sequence, respectively). During intracerebral recordings, highly variable natural images of faces were inserted periodically every fifth image (i.e., face-selective frequency at 1.2 Hz = 6/5 Hz) as usually performed (*Figure 1B*; *Rossion et al., 2015*; *Jonas et al., 2016*). A stimulation sequence lasted 70 s, including 66 s of stimulation at full contrast flanked by 2 s of fade-in and fade-out, with gradual increases and decreases in contrast, respectively. All images were presented with a sinusoidal stimulation contrast to provide a smooth transition between successive images. Participants were unaware of the periodicity of the faces. During both fMRI and intracerebral recordings, participants stared at a small black cross presented at the center of the stimuli and detected rare brief nonperiodic color changes (70–10 times per sequence for fMRI/iEEG respectively, for 500 ms) of the fixation cross (black to red). The experiments were conducted using MATLAB for P1 and Java 8 for P2.

## Electrophysiology: spike sorting

Spike detection and sorting/clustering were carried out using an automatic algorithm, based on a Bayesian approach (*Le Cam et al., 2023*). Neurons were classified into clusters based on (1) non-causal band-pass (300–6000 Hz) filtering, (2) Median Absolute Deviation-based spike detection, and (3) artifact removal consisting of removing bounces and events common to more than 3 microwires. The sorted clusters were visually reviewed and classified into single-units (SU) or multi-units (MU) based on their spike shape, variance, inter-spike interval distribution, and the presence of a refractory period. In all the sessions held by the two participants, 299 SU and 62 MU were identified. Across all independent sessions (i.e., recorded on different half-days), 206 SU and 39 MU were retained for analyses (P1: $N = 62$ SU, $N = 13$ MU; P2: $N = 144$ SU, $N = 26$ MU). An average of 2 units per contact was isolated.

## Electrophysiology: frequency-domain and time-domain analyses

Analyses were performed using the free software *Letswave 6,* running on MATLAB R2022a. Signal processing and frequency-domain analyses were similar to previous SEEG studies (*Jonas et al., 2016*;

*Hagen et al., 2020*), except they were applied to the sorted MU/SU raster trains rather than raw SEEG voltage fluctuations. A discrete Fast Fourier Transform (FFT) was applied to the spike trains, and the resulting amplitude spectrum was cut into 1 Hz segments centered on the face-selective frequency and its three additional harmonics (i.e., 1.2 Hz, 2.4 Hz, 3.6 Hz, and 4.8 Hz), as well as on the base stimulation frequency and its harmonics (i.e., 6 Hz and 12 Hz). The amplitude of these FFT segments was summed and transformed into a Z-score computed as the difference between the amplitude at the target frequency bin and the mean amplitude of 20 surrounding bins (10 on each side) divided by the standard deviation of amplitudes in the corresponding 20 bins. SNR spectra were also calculated as the ratio between the amplitude at each frequency bin and the average of the corresponding 20 surrounding bins (11 on each side, excluding the 2 bins directly adjacent to the bin of interest). A cluster (SU or MU) was considered as showing a significant response for faces and classified as 'face-selective' if the Z-score at the target frequency bin exceeded 1.64 (i.e., $p < 0.05$ one-tailed). To isolate face-selective responses from responses to non-face objects in the time domain (as in *Figure 2C and F*), a FFT notch filter (filter width = 0.05 Hz) was then applied to the 70 s single or multi-units spike trains to remove the general visual response at 6 Hz and two additional harmonics (i.e., 12 and 18 Hz). To account for the sinusoidal modulation of contrast, the face onset time was shifted forward by 33 ms (~1/5 of a 6 Hz cycle duration). This delay was estimated by comparing SEEG responses to sequences presented with sinewave or squarewave (i.e., abrupt) contrast modulation of visual stimulation. The onset of face-selective response was delayed by 30–35 ms for sinusoidal visual stimulation, which corresponds to 4 screen refresh frames (33 ms) and 35% of the full contrast. The spike trains of each identified cluster were then segmented into 1 s segments around face onset, and the resulting epochs were temporally smoothed (20 ms time window) and averaged. The net average spike rate was calculated by subtracting each sequence's mean baseline spike rate in a [-0.166–0 s] time window relative to face onset. Finally, the latency was computed as the time point at which net firing crossed the baseline +/-2.58 s.d. value (i.e., $p < 0.01$, two-tailed percentile bootstrap) for at least 30 ms, as described in *Jacques et al., 2022*.

## fMRI acquisition

The two participants were scanned at the CHRU-Nancy, with a 3T Siemens Magnetom Prisma system (Siemens Medical System, Erlangen, Germany) with a 64-channel head-neck coil. Anatomical images were collected using a high-resolution T1-weighted magnetization-prepared gradient-echo image (MP-RAGE) sequence (192 sagittal slices, TR = 2300 ms, TE = 2.6 ms; flip angle (FA)=9°, field of view (FOV) = 256 × 256). Functional images were collected with a T2*-weighted simultaneous multi-slice echo planar imaging (SMS EPI) sequence (TR = 1500 ms, TE = 30 ms, FA = 72°, FOV = 240 × 240 mm², voxel size = 2.5 mm isotropic, matrix size = 96 × 96 for sequences done in 2019 or matrix size = 80 × 80 for sequences done in 2022, interleaved), which acquired 44 oblique-axial slices covering the entire temporal and occipital lobes. The total duration of each sequence (run) for P1 (tested in 2019) was 333 s, including 9 s of dummy scans (222 TRs). The total duration of each run for P2 (tested in 2022) was 405 s including 9 s of dummy scans (270 TRs). Images were back-projected onto a projection screen by an MRI-compatible LCD projector. The participants observed the sequences through a mirror placed within the FR head coil. The images subtended a viewing angle of 8° × 8° (33.4 cm × 33.4 cm) at a viewing distance of 240 cm. Three fMRI sequences were performed for each patient, who was scanned 6–8 weeks after the microelectrode recordings.

## fMRI analyses

As in *Laurent et al., 2023*, the volumes of each run were first rigidly realigned with each other, and a mean functional image of the runs was computed for co-registration with the anatomical image, using *SPM12*. The volumes were also spatially smoothed with a Gaussian kernel of 2 mm (FWHM; i.e., full width at half maximum). The functional runs were averaged across runs in the volumetric space. Then, a Fourier analysis was performed using the FFT function in MATLAB without windowing. The FFT was applied to the entire BOLD response time course, and the amplitude spectrum was directly derived from the Fourier transform coefficients. Amplitudes at the face stimulation frequency (i.e., at $F = 0.111$ Hz) were converted into Z-scores, using the mean and standard deviation of the amplitude at neighboring frequencies (see *Gao et al., 2018* for details). The relative activations and deactivations of the neural responses at the face stimulation frequency were defined by the phase of the BOLD

response. For each individual, the histogram of phase values (20 bins) of all the voxels with a Z-score > 3.1 and with a positive phase value was calculated. The phase value of the histogram bin that has the largest number was used as the center phase ($\varphi$) to define all the voxels with their phase values within $\varphi \pm \pi/2$ as activations (+sign) and voxels with their phase values outside of this window as deactivations ($-$ sign). These signs were then applied to Z-score maps to obtain the final response map.

## Acknowledgements

We thank the two participants for their involvement in the study. This research was supported by the ERC Adg HumanFace 101055175 awarded to Bruno Rossion, a grant from "Agence Nationale de la Recherche" (ANR-23-CE37-0016, ANR PREFER) awarded to Bruno Rossion and Benoit R Cottereau and a PhD fellowship from the Université de Lorraine to Marie-Alphée Laurent.

## Additional information

### Funding

| Funder | Grant reference number | Author |
|---|---|---|
| European Research Council | 10.3030/101055175 | Bruno Rossion |
| Agence Nationale de la Recherche | ANR-23-CE37-0016 | Benoit R Cottereau Bruno Rossion |
| Université de Lorraine | PhD fellowship | Marie-Alphée Laurent |

The funders had no role in study design, data collection and interpretation, or the decision to submit the work for publication.

### Author contributions

Marie-Alphée Laurent, Conceptualization, Data curation, Formal analysis, Investigation, Visualization, Methodology, Writing – original draft, Writing – review and editing; Corentin Jacques, Formal analysis, Investigation, Visualization, Methodology, Writing – review and editing; Xiaoqian Yan, Formal analysis, Visualization, Methodology; Pauline Jurczynski, Software, Formal analysis, Visualization, Methodology; Sophie Colnat-Coulbois, Investigation; Louis Maillard, Laurent Koessler, Resources, Funding acquisition, Project administration; Steven Le Cam, Radu Ranta, Software; Benoit R Cottereau, Supervision, Visualization, Writing – review and editing; Jacques Jonas, Conceptualization, Supervision, Validation, Investigation, Methodology; Bruno Rossion, Conceptualization, Supervision, Funding acquisition, Validation, Investigation, Visualization, Methodology, Writing – original draft, Writing – review and editing

### Author ORCIDs

Marie-Alphée Laurent ⓘ https://orcid.org/0000-0001-6585-7257
Benoit R Cottereau ⓘ https://orcid.org/0000-0002-2624-7680
Bruno Rossion ⓘ https://orcid.org/0000-0002-1845-3935

### Ethics

Human subjects: The two participants gave written informed consent to participate in the study (REUNIE, 2015-A01951-48), which was approved by a national ethical committee (CPP Est III, No. 16.02.01).

Reviewer #1 (Public review): https://doi.org/10.7554/eLife.104779.3.sa1
Reviewer #2 (Public review): https://doi.org/10.7554/eLife.104779.3.sa2
Reviewer #3 (Public review): https://doi.org/10.7554/eLife.104779.3.sa3
Author response https://doi.org/10.7554/eLife.104779.3.sa4

## Additional files

### Supplementary files
MDAR checklist

### Data availability
Intracerebral EEG and fMRI data are accessible on OSF.

The following dataset was generated:

| Author(s) | Year | Dataset title | Dataset URL | Database and Identifier |
|---|---|---|---|---|
| Laurent MA, Jacques C, Yan X, Jurczynski P, Colnat-Coulbois S, Maillard L, Cam SL, Ranta R, Cottereau B, Koessler L, Jonas J, Rossion B | 2024 | A tight relationship between BOLD fMRI activation/deactivation and increase/decrease in single neuron responses in human association cortex | https://osf.io/hnafv | Open Science Framework, hnafv |

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
