## [Editor Report · eLife Assessment]

This **valuable** short paper is an ingenious use of clinical patient data to address an issue in imaging neuroscience. The authors clarify the role of face-selectivity in human fusiform gyrus by measuring both BOLD fMRI and depth electrode recordings in the same individuals; furthermore, by comparing responses in different brain regions in the two patients, they suggested that the suppression of blood oxygenation is associated with a decrease in local neural activity. The methods are **solid** and provide a rare dataset of potentially general importance.

---

## [Referee Report · Reviewer #1 (Public review)]

Summary:

Measurement of BOLD MR imaging has regularly found regions of the brain that show reliable suppression of BOLD responses during specific experimental testing conditions. These observations are to some degree unexplained, in comparison with more usual association between activation of the BOLD response and excitatory activation of the neurons (most tightly linked to synaptic activity) in the same brain location. This paper finds two patients whose brains were tested with both non-invasive functional MRI and with invasive insertion of electrodes, which allowed the direct recording of neuronal activity. The electrode insertions were made within the fusiform gyrus, which is known to process information abouit faces, in a clinical search for the sites of intractable epilepsy in each patient. The simple observation is that the electrode location in one patient showed activation of the BOLD response and activation of neuronal firing in response to face stimuli. This is the classical association. The other patient showed an informative and different pattern of responses. In this person, the electrode location showed a suppression of the BOLD response to face stimuli and, most interestingly, an associated suppression of neuronal activity at the electrode site.

Strengths:

Whilst these results are not by themselves definitive, they add an important piece of evidence to a long-standing discussion about the origins of the BOLD response. The observation of decreased neuronal activation associated with negative BOLD is interesting because, at various times, exactly the opposite association has been predicted. It has been previously argued that if synaptic mechanisms of neuronal inhibition are responsible for the suppression of neuronal firing, then it would be reasonable

Weaknesses:

The chief weakness of the paper is that the results may be unique in a slightly awkward way. The observation of positive BOLD and neuronal activation is made at one brain site in one patient, while the complementary observation of negative BOLD and neuronal suppression actually derives from the other patient. Showing both effects in both patients would make a much stronger paper.

Comments on revisions:

The material on lines 165-175 should not be left hidden away in the Methods section. This should be highlighted in the Discussion as a limitation of the current study and an issue that could be improved upon in future studies.

---

## [Referee Report · Reviewer #2 (Public review)]

Summary:

This is a short and straightforward paper describing BOLD fMRI and depth electrode measurements from two regions of the fusiform gyrus that show either higher or lower BOLD responses to faces vs. objects (which I will call face-positive and face-negative regions). In these regions, which were studied separately in two patients undergoing epilepsy surgery, spiking activity increased for faces relative to objects in the face-positive region and decreased for faces relative to objects in the face-negative region. Interestingly, about 30% of neurons in the face-negative region did not respond to objects and decreased their responses below baseline in response to faces (absolute suppression).

Strengths:

These patient data are valuable, with many recording sessions and neurons from human face-selective regions, and the methods used for comparing face and object responses in both fMRI and electrode recordings were robust and well-established. The finding of absolute suppression could clarify the nature of face selectivity in human fusiform gyrus, since previous fMRI studies of the face-negative region could not distinguish whether face < object responses came from absolute suppression, or just relatively lower but still positive responses to faces vs. objects.

Weaknesses:

The authors claim that the results tell us about both (1) face-selectivity in the fusiform gyrus, and (2) the physiological basis of the BOLD signal. However, I would like to see more of the data that supports the first claim included in the paper.

The authors report that ~30% of neurons showed absolute suppression, but those data are not shown separately from the neurons that only show relative reductions. It is difficult to evaluate the absolute suppression claim from the short assertion in the text alone (lines 105-106), although this is a critical claim in the paper.

Comments on revisions:

The authors have provided a figure showing one example neuron that shows absolute suppression in their response to reviewers; I would recommend including a similar panel in one of the paper figures showing data averaged across all neurons classified as showing absolute suppression.

---

## [Referee Report · Reviewer #3 (Public review)]

Summary:

In this paper the authors conduct two experiments an fMRI experiment and intracranial recordings of neurons in two patients P1 and P2. In both experiments, they employ a SSVEP paradigm in which they show images at a fast rate (e.g. 6Hz) and then they show face images at a slower rate (e.g. 1.2Hz), where the rest of the images are a variety of object images. In the first patient, they record from neurons over a region in the mid fusiform gyrus that is face-selective and in the second patient, they record neurons from a region more medially that is not face selective (it responds more strongly to objects than faces). Results find similar selectivity between the electrophysiology data and the fMRI data in that the location which shows higher fMRI to faces also finds face-selective neurons and the location which finds preference to non faces also shows non face preferring neurons.

Strengths:

The data is important in that it shows that there is a relationship between category selectivity measured from electrophysiology data and category-selective from fMRI. The data is unique as it contains a lot of single and multiunit recordings (245 units) from the human fusiform gyrus - which the authors point out - is a humanoid specific gyrus.

Weaknesses:

My major concerns are two-fold:(i) There is a paucity of data; Thus, more information (results and methods) is warranted; and in particular there is no comparison between the fMRI data and the SEEG data.

(ii) One main claim of the paper is that there is evidence for suppressed responses to faces in the non-face selective region. That is, the reduction in activation to faces in the non-face selective region is interpreted as a suppression in the neural response and consequently the reduction in fMRI signal is interpreted as suppression. However, the SSVEP paradigm has no baseline (it alternates between faces and objects) and therefore it cannot distinguish between lower firing rate to faces vs suppression of response to faces.

(1) Additional data: the paper has 2 figures: figure 1 which shows the experimental design and figure 2 which presents data, the latter shows one example neuron raster plot from each patient and group average neural data from each patient. In this reader's opinion this is insufficient data to support the conclusions of the paper. The paper will be more impactful if the researchers would report the data more comprehensively.

(a) There is no direct comparison between the fMRI data and the SEEG data, except for a comparison of the location of the electrodes relative to the statistical parametric map generated from a contrast (Fig 2a,d). It will be helpful to build a model linking between the neural responses to the voxel response in the same location - i.e., estimate from the electrophysiology data the fMRI data (e.g. Logothetis & Wandell, 2004)

(b) More comprehensive analyses of the SSVEP neural data: It will be helpful to show the results of the frequency analyses of the SSVEP data for all neurons to show that there are significant visual responses and significant face responses. It will be also useful to compare and quantify the magnitude of the face responses compared to the visual responses.

(c) The neuron shown in E shows cyclical responses tied to the onset of the stimuli, is this the visual response? If so, why is there an increase in the firing rate of the neuron before the face stimulus is shown in time 0? The neuron's data seems different than the average response across neurons; This raises a concern about interpreting the average response across neurons in panel F which seems different than the single neuron responses

(d) Related to (c) it would be useful to show raster plots of all neurons and quantify if the neural responses within a region are homogeneous or heterogeneous. This would add data relating the single neuron response to the population responses measured from fMRI. See also Nir 2009.

(e) When reporting group average data (e.g., Fig 2C,F) it is necessary to show standard deviation of the response across neurons.

(f) Is it possible to estimate the latency of the neural responses to face and object images from the phase data? If so, this will add important information on the timing of neural responses in the human fusiform gyrus to face and object images.

(g) Related to (e) In total the authors recorded data from 245 units (some single units and some multiunits) and they found that both in the face and nonface selective most of the recoded neurons exhibited face -selectivity, which this reader found confusing: They write " Among all visually responsive neurons, we 87 found a very high proportion of face-selective neurons (p < 0.05) in both activated 88 and deactivated MidFG regions (P1: 98.1%; N = 51/52; P2: 86.6%; N = 110/127)'. Is the face selectivity in P1 an increase in response to faces and P2 a reduction in response to faces or in both it's an increase in response to faces

(1) Additional methods(a) it is unclear if the SSVEP analyses of neural responses were done on the spikes or the raw electrical signal. If the former, how is the SSVEP frequency analysis done on discrete data like action potentials?(b) it is unclear why the onset time was shifted by 33ms; one can measure the phase of the response relative to the cycle onset and use that to estimate the delay between the onset of a stimulus and the onset of the response. Adding phase information will be useful.

(2) Interpretation of suppression:

The SSVEP paradigm alternates between 2 conditions: faces and objects and has no baseline; In other words, responses to faces are measured relative to the baseline response to objects so that any region that contains neurons that have a lower firing rate to faces than objects is bound to show a lower response in the SSVEP signal. Therefore, because the experiment does not have a true baseline (e.g. blank screen, with no visual stimulation) this experimental design cannot distinguish between lower firing rate to faces vs suppression of response to faces.The strongest evidence put forward for suppression is the response of non-visual neurons that was also reduced when patients looked at faces, but since these are non-visual neurons, it is unclear how to interpret the responses to faces.

Comments on revisions:

In the revision, the authors added information and answered several of the main questions. Several points remain unanswered because the authors would like to publish a short format paper here, and suggest that answering these questions is outside the scope of the paper. The authors would like to leave some of the more detailed analyses for a subsequent longer paper.

---

## [Author Response]

The following is the authors’ response to the original reviews.

**Reviewer #1 (Public review):**
Summary:Measurement of BOLD MR imaging has regularly found regions of the brain that show reliable suppression of BOLD responses during specific experimental testing conditions. These observations are to some degree unexplained, in comparison with more usual association between activation of the BOLD response and excitatory activation of the neurons (most tightly linked to synaptic activity) in the same brain location. This paper finds two patients whose brains were tested with both non-invasive functional MRI and with invasive insertion of electrodes, which allowed the direct recording of neuronal activity. The electrode insertions were made within the fusiform gyrus, which is known to process information about faces, in a clinical search for the sites of intractable epilepsy in each patient. The simple observation is that the electrode location in one patient showed activation of the BOLD response and activation of neuronal firing in response to face stimuli. This is the classical association. The other patient showed an informative and different pattern of responses. In this person, the electrode location showed a suppression of the BOLD response to face stimuli and, most interestingly, an associated suppression of neuronal activity at the electrode site.Strengths:Whilst these results are not by themselves definitive, they add an important piece of evidence to a long-standing discussion about the origins of the BOLD response. The observation of decreased neuronal activation associated with negative BOLD is interesting because, at various times, exactly the opposite association has been predicted. It has been previously argued that if synaptic mechanisms of neuronal inhibition are responsible for the suppression of neuronal firing, then it would be reasonableWeaknesses:The chief weakness of the paper is that the results may be unique in a slightly awkward way. The observation of positive BOLD and neuronal activation is made at one brain site in one patient, while the complementary observation of negative BOLD and neuronal suppression actually derives from the other patient. Showing both effects in both patients would make a much stronger paper.

We thank reviewer #1 for their positive evaluation of our paper. Obviously, we agree with the reviewer that the paper would be much stronger if BOTH effects – spike increase and decrease – would be found in BOTH patients in their corresponding fMRI regions (lateral and medial fusiform gyrus) (also in the same hemisphere). Nevertheless, we clearly acknowledge this limitation in the (revised) version of the manuscript (p.8: Material and Methods section).

Note that with respect to the fMRI data, our results are not surprising, as we indicate in the manuscript: BOLD increases to faces (relative to nonface objects) are typically found in the LatFG and BOLD decreases in the medialFG in the revised version, we have added the reference to an early neuroimaging paper that describes this dissociation clearly:

Pelphrey, K. A., Mack, P. B., Song, A., Güzeldere, G., & McCarthy, G. Faces evoke spatially differentiated patterns of BOLD activation and deactivation. Neuroreport 14, 955–959 (2003).

This pattern of increase/decrease in fMRI can be appreciated in both patients on Figure 2, although one has to consider both the transverse and coronal slices to appreciate it.

Regarding electrophysiological data, in the current paper, one could think that P1 shows only increases to faces, and P2 would show only decreases (irrespective of the region). However, that is not the case since 11% of P1’s face-selective units are decreases (89% are increases) and 4% of P2’s face-selective units are increases. This has now been made clearer in the revised manuscript (p.5).

As the reviewer is certainly aware, the number and positions of the electrodes are based on strict clinical criteria, and we will probably never encounter a situation with two neighboring (macro-micro hybrid electrodes), one with microelectrodes ending up in the lateral MidFG, the other in the medial MidFG, in the same patient. If there is no clinical value for the patient, this cannot be done.

The only thing we can do is to strengthen these results in the future by collecting data on additional patients with an electrode either in the lateral or the medial FG, together with fMRI. But these are the only two patients we have been able to record so far with electrodes falling unambiguously in such contrasted regions and with large (and comparable) measures.

While we acknowledge that the results may be unique because of the use of 2 contrasted patients only (and this is why the paper is a short report), the data is compelling in these 2 cases, and we are confident that it will be replicated in larger cohorts in the future.

Finally, information regarding ethics approval has been provided in the paper.

**Reviewer #2 (Public review):**
Summary:This is a short and straightforward paper describing BOLD fMRI and depth electrode measurements from two regions of the fusiform gyrus that show either higher or lower BOLD responses to faces vs. objects (which I will call face-positive and facenegative regions). In these regions, which were studied separately in two patients undergoing epilepsy surgery, spiking activity increased for faces relative to objects in the face-positive region and decreased for faces relative to objects in the face-negative region. Interestingly, about 30% of neurons in the face-negative region did not respond to objects and decreased their responses below baseline in response to faces (absolute suppression).Strengths:These patient data are valuable, with many recording sessions and neurons from human face-selective regions, and the methods used for comparing face and object responses in both fMRI and electrode recordings were robust and well-established. The finding of absolute suppression could clarify the nature of face selectivity in human fusiform gyrus since previous fMRI studies of the face-negative region could not distinguish whether face < object responses came from absolute suppression, or just relatively lower but still positive responses to faces vs. objects.Weaknesses:The authors claim that the results tell us about both (1) face-selectivity in the fusiform gyrus, and (2) the physiological basis of the BOLD signal. However, I would like to see more of the data that supports the first claim, and I am not sure the second claim is supported.(1) The authors report that ~30% of neurons showed absolute suppression, but those data are not shown separately from the neurons that only show relative reductions. It is difficult to evaluate the absolute suppression claim from the short assertion in the text alone (lines 105-106), although this is a critical claim in the paper.

We thank reviewer #2 for their positive evaluation of our paper. We understand the reviewer’s point, and we partly agree. Where we respectfully disagree is that the finding of absolute suppression is critical for the claim of the paper: finding an identical contrast between the two regions in terms of RELATIVE increase/decrease of face-selective activity in fMRI and spiking activity is already novel and informative. Where we agree with the reviewer is that the absolute suppression could be more documented: it wasn’t, due to space constraints (brief report). We provide below an example of a neuron showing absolute suppression to faces (P2), as also requested in the recommendations to authors. In the frequency domain, there is only a face-selective response (1.2 Hz and harmonics) but no significant response at 6 Hz (common general visual response). In the time-domain, relative to face onset, the response drops below baseline level. It means that this neuron has baseline (non-periodic) spontaneous spiking activity that is actively suppressed when a face appears.

**Author response image 1. sa4fig1:** 

(2) I am not sure how much light the results shed on the physiological basis of the BOLD signal. The authors write that the results reveal "that BOLD decreases can be due to relative, but also absolute, spike suppression in the human brain" (line 120). But I think to make this claim, you would need a region that exclusively had neurons showing absolute suppression, not a region with a mix of neurons, some showing absolute suppression and some showing relative suppression, as here. The responses of both groups of neurons contribute to the measured BOLD signal, so it seems impossible to tell from these data how absolute suppression per se drives the BOLD response.

It is a fact that we find both kinds of responses in the same region. We cannot tell with this technique if neurons showing relative vs. absolute suppression of responses are spatially segregated for instance (e.g., forming two separate sub-regions) or are intermingled. And we cannot tell from our data how absolute suppression per se drives the BOLD response. In our view, this does not diminish the interest and originality of the study, but the statement "that BOLD decreases can be due to relative, but also absolute, spike suppression in the human brain” has been rephrased in the revised manuscript: "that BOLD decreases can be due to relative, or absolute (or a combination of both), spike suppression in the human brain”.

**Reviewer #3 (Public review):**
In this paper the authors conduct two experiments an fMRI experiment and intracranial recordings of neurons in two patients P1 and P2. In both experiments, they employ a SSVEP paradigm in which they show images at a fast rate (e.g. 6Hz) and then they show face images at a slower rate (e.g. 1.2Hz), where the rest of the images are a variety of object images. In the first patient, they record from neurons over a region in the mid fusiform gyrus that is face-selective and in the second patient, they record neurons from a region more medially that is not face selective (it responds more strongly to objects than faces). Results find similar selectivity between the electrophysiology data and the fMRI data in that the location which shows higher fMRI to faces also finds face-selective neurons and the location which finds preference to non faces also shows non face preferring neurons.Strengths:The data is important in that it shows that there is a relationship between category selectivity measured from electrophysiology data and category-selective from fMRI. The data is unique as it contains a lot of single and multiunit recordings (245 units) from the human fusiform gyrus - which the authors point out - is a humanoid specific gyrus.Weaknesses:My major concerns are two-fold:(i) There is a paucity of data; Thus, more information (results and methods) is warranted; and in particular there is no comparison between the fMRI data and the SEEG data.

We thank reviewer #3 for their positive evaluation of our paper. If the reviewer means paucity of data presentation, we agree and we provide more presentation below, although the methods and results information appear as complete to us. The comparison between fMRI and SEEG is there, but can only be indirect (i.e., collected at different times and not related on a trial-by-trial basis for instance). In addition, our manuscript aims at providing a short empirical contribution to further our understanding of the relationship between neural responses and BOLD signal, not to provide a model of neurovascular coupling.

(ii) One main claim of the paper is that there is evidence for suppressed responses to faces in the non-face selective region. That is, the reduction in activation to faces in the non-face selective region is interpreted as a suppression in the neural response and consequently the reduction in fMRI signal is interpreted as suppression. However, the SSVEP paradigm has no baseline (it alternates between faces and objects) and therefore it cannot distinguish between lower firing rate to faces vs suppression of response to faces.

We understand the concern of the reviewer, but we respectfully disagree that our paradigm cannot distinguish between lower firing rate to faces vs. suppression of response to faces. Indeed, since the stimuli are presented periodically (6 Hz), we can objectively distinguish stimulus-related activity from spontaneous neuronal firing. The baseline corresponds to spikes that are non-periodic, i.e., unrelated to the (common face and object) stimulation. For a subset of neurons, even this non-periodic baseline activity is suppressed, above and beyond the suppression of the 6 Hz response illustrated on Figure 2. We mention it in the manuscript, but we agree that we do not present illustrations of such decrease in the time-domain for SU, which we did not consider as being necessary initially (please see below for such presentation).

(1) Additional data: the paper has 2 figures: figure 1 which shows the experimental design and figure 2 which presents data, the latter shows one example neuron raster plot from each patient and group average neural data from each patient. In this reader's opinion this is insufficient data to support the conclusions of the paper. The paper will be more impactful if the researchers would report the data more comprehensively.

We answer to more specific requests for additional evidence below, but the reviewer should be aware that this is a short report, which reaches the word limit. In our view, the group average neural data should be sufficient to support the conclusions, and the example neurons are there for illustration. And while we cannot provide the raster plots for a large number of neurons, the anonymized data is made available at:

(a) There is no direct comparison between the fMRI data and the SEEG data, except for a comparison of the location of the electrodes relative to the statistical parametric map generated from a contrast (Fig 2a,d). It will be helpful to build a model linking between the neural responses to the voxel response in the same location - i.e., estimate from the electrophysiology data the fMRI data (e.g., Logothetis & Wandell, 2004).

As mentioned above the comparison between fMRI and SEEG is indirect (i.e., collected at different times and not related on a trial-by-trial basis for instance) and would not allow to make such a model.

(b) More comprehensive analyses of the SSVEP neural data: It will be helpful to show the results of the frequency analyses of the SSVEP data for all neurons to show that there are significant visual responses and significant face responses. It will be also useful to compare and quantify the magnitude of the face responses compared to the visual responses.

The data has been analyzed comprehensively, but we would not be able to show all neurons with such significant visual responses and face-selective responses.

(c) The neuron shown in E shows cyclical responses tied to the onset of the stimuli, is this the visual response?

Correct, it’s the visual response at 6 Hz.

If so, why is there an increase in the firing rate of the neuron before the face stimulus is shown in time 0?

Because the stimulation is continuous. What is displayed at 0 is the onset of the face stimulus, with each face stimulus being preceded by 4 images of nonface objects.

The neuron's data seems different than the average response across neurons; This raises a concern about interpreting the average response across neurons in panel F which seems different than the single neuron responses

The reviewer is correct, and we apologize for the confusion. This is because the average data on panel F has been notch-filtered for the 6 Hz (and harmonic responses), as indicated in the methods (p.11): ‘a FFT notch filter (filter width = 0.05 Hz) was then applied on the 70 s single or multi-units time-series to remove the general visual response at 6 Hz and two additional harmonics (i.e., 12 and 18 Hz)’.

Here is the same data without the notch-filter (the 6Hz periodic response is clearly visible):

**Author response image 2. sa4fig2:** 

For sake of clarity, we prefer presenting the notch-filtered data in the paper, but the revised version makes it clear in the figure caption that the average data has been notch-filtered.

(d) Related to (c) it would be useful to show raster plots of all neurons and quantify if the neural responses within a region are homogeneous or heterogeneous. This would add data relating the single neuron response to the population responses measured from fMRI. See also Nir 2009.

We agree with the reviewer that this is interesting, but again we do not think that it is necessary for the point made in the present paper. Responses in these regions appear rather heterogenous, and we are currently working on a longer paper with additional SEEG data (other patients tested for shorter sessions) to define and quantify the face-selective neurons in the MidFusiform gyrus with this approach (without relating it to the fMRI contrast as reported here).

(e) When reporting group average data (e.g., Fig 2C,F) it is necessary to show standard deviation of the response across neurons.

We agree with the reviewer and have modified Figure 2 accordingly in the revised manuscript.

(f) Is it possible to estimate the latency of the neural responses to face and object images from the phase data? If so, this will add important information on the timing of neural responses in the human fusiform gyrus to face and object images.

The fast periodic paradigm to measure neural face-selectivity has been used in tens of studies since its original reports:

in EEG: Rossion et al., 2015: https://doi.org/10.1167/15.1.18in SEEG: Jonas et al., 2016: https://doi.org/10.1073/pnas.1522033113

In this paradigm, the face-selective response spreads to several harmonics (1.2 Hz, 2.4 Hz, 3.6 Hz, etc.) (which are summed for quantifying the total face-selective amplitude). This is illustrated below by the averaged single units’ SNR spectra across all recording sessions for both participants.

**Author response image 3. sa4fig3:** 

There is no unique phase-value, each harmonic being associated with a phase-value, so that the timing cannot be unambiguously extracted from phase values. Instead, the onset latency is computed directly from the time-domain responses, which is more straightforward and reliable than using the phase. Note that the present paper is not about the specific time-courses of the different types of neurons, which would require a more comprehensive report, but which is not necessary to support the point made in the present paper about the SEEG-fMRI sign relationship.

(g) Related to (e) In total the authors recorded data from 245 units (some single units and some multiunits) and they found that both in the face and nonface selective most of the recoded neurons exhibited face -selectivity, which this reader found confusing: They write “ Among all visually responsive neurons, we found a very high proportion of face-selective neurons (p < 0.05) in both activated and deactivated MidFG regions (P1: 98.1%; N = 51/52; P2: 86.6%; N = 110/127)’. Is the face selectivity in P1 an increase in response to faces and P2 a reduction in response to faces or in both it’s an increase in response to faces

Face-selectivity is defined as a DIFFERENTIAL response to faces compared to objects, not necessarily a larger response to faces. So yes, face-selectivity in P1 is an increase in response to faces and P2 a reduction in response to faces.

Additional methods(a) it is unclear if the SSVEP analyses of neural responses were done on the spikes or the raw electrical signal. If the former, how is the SSVEP frequency analysis done on discrete data like action potentials?

The FFT is applied directly on spike trains using Matlab’s discrete Fourier Transform function. This function is suitable to be applied to spike trains in the same way as to any sampled digital signal (here, the microwires signal was sampled at 30 kHz, see Methods).

In complementary analyses, we also attempted to apply the FFT on spike trains that had been temporally smoothed by convolving them with a 20ms square window (Le Cam et al., 2023, cited in the paper). This did not change the outcome of the frequency analyses in the frequency range we are interested in. We have also added one sentence with information in the methods section about spike detection (p.10).

(b) it is unclear why the onset time was shifted by 33ms; one can measure the phase of the response relative to the cycle onset and use that to estimate the delay between the onset of a stimulus and the onset of the response. Adding phase information will be useful.

The onset time was shifted by 33ms because the stimuli are presented with a sinewave contrast modulation (i.e., at 0ms, the stimulus has 0% contrast). 100% contrast is reached at half a stimulation cycle, which is 83.33ms here, but a response is likely triggered before reaching 100% contrast. To estimate the delay between the start of the sinewave (0% contrast) and the triggering of a neural response, we tested 7 SEEG participants with the same images presented in FPVS sequences either as a sinewave contrast (black line) modulation or as a squarewave (i.e. abrupt) contrast modulation (red line). The 33ms value is based on these LFP data obtained in response to such sinewave stimulation and squarewave stimulation of the same paradigm. This delay corresponds to 4 screen refresh frames (120 Hz refresh rate = 8.33ms by frame) and 35% of the full contrast, as illustrated below (please see also Retter, T. L., & Rossion, B. (2016). Uncovering the neural magnitude and spatio-temporal dynamics of natural image categorization in a fast visual stream. Neuropsychologia, 91, 9–28).

**Author response image 4. sa4fig4:** 

(2) Interpretation of suppression:The SSVEP paradigm alternates between 2 conditions: faces and objects and has no baseline; In other words, responses to faces are measured relative to the baseline response to objects so that any region that contains neurons that have a lower firing rate to faces than objects is bound to show a lower response in the SSVEP signal. Therefore, because the experiment does not have a true baseline (e.g. blank screen, with no visual stimulation) this experimental design cannot distinguish between lower firing rate to faces vs suppression of response to faces.The strongest evidence put forward for suppression is the response of non-visual neurons that was also reduced when patients looked at faces, but since these are non-visual neurons, it is unclear how to interpret the responses to faces.

We understand this point, but how does the reviewer know that these are non-visual neurons? Because these neurons are located in the visual cortex, they are likely to be visual neurons that are not responsive to non-face objects. In any case, as the reviewer writes, we think it’s strong evidence for suppression.

We thank all three reviewers for their positive evaluation of our paper and their constructive comments.